# Intake of Low Glycaemic Index Foods but Not Probiotics Is Associated with Atherosclerosis Risk in Women with Polycystic Ovary Syndrome

**DOI:** 10.3390/life13030799

**Published:** 2023-03-15

**Authors:** Aleksandra Bykowska-Derda, Małgorzata Kałużna, Agnieszka Garbacz, Katarzyna Ziemnicka, Marek Ruchała, Magdalena Czlapka-Matyasik

**Affiliations:** 1Department of Human Nutrition and Dietetics, Poznan University of Life Sciences, Wojska Polskiego 31, 60-624 Poznan, Poland; aleksandra.derda@up.poznan.pl; 2Department of Endocrinology, Metabolism and Internal Diseases, Poznan University of Medical Sciences, 60-355 Poznan, Poland; 3Student Science Club of Dieticians, Poznan University of Life Sciences, Wojska Polskiego 31, 60-624 Poznan, Poland

**Keywords:** atherosclerosis, frequency intake, diet quality, BodPod, body composition, AIP

## Abstract

Women with polycystic ovary syndrome (PCOS) are at high cardiometabolic risk. The atherogenic index of plasma (AIP) strongly predicts atherosclerosis. Some studies suggest that probiotic intake may lower AIP. This study analysed the relationship between the frequency of dietary intake of low glycaemic index (prebiotic) and probiotic foods and atherosclerosis risk in women with PCOS. Methods: A total of 127 women were divided into two groups: AIP over 0.11 (highAIP) and AIP ≤ 0.11 (lowAIP). The KomPAN^®^ questionnaire was used to measure food frequency intake; pro-healthy, non-healthy, low glycaemic and probiotic dietary indexes were calculated based on daily food consumption. Body composition was measured by air displacement plethysmography (BodPod). AIP was calculated as a logarithm of triglycerides and high-density lipoproteins from plasma. Results: The highAIP group was 63% less likely to consume low glycaemic index foods three or more times a day than the lowAIP group. The HighAIP group was also 62% less likely to consume buckwheat, oats, whole-grain pasta or coarse-ground grains at least a few times a week. Pro-healthy foods tended to be less frequently consumed by the highAIP group, when adjusted for BMI and age. Conclusion: Women with PCOS at high risk of atherosclerosis consumed less low glycaemic index foods than women with a low risk of atherosclerosis. Intake of high-fibre, low glycaemic index foods could prevent atherosclerosis in women with PCOS; however, the effect of probiotic food intake remains unclear.

## 1. Introduction

Polycystic ovary syndrome (PCOS) is a heterogeneous disorder that affects reproductive-age women, and is characterised by both ovarian cysts and high levels of androgenic hormones [1]. This disorder can be accompanied by insulin sensitivity, diabetes, high blood pressure, hypercholesterolemia, and depression [2]. Additionally, more than 50% of women with PCOS are believed to be overweight or obese, with an increased cardiometabolic risk due to the conditions listed above [3,4,5].

Research has shown that the vast majority of women with PCOS consume an unbalanced diet deficient in fibre, fatty acid omega 3, calcium, magnesium, zinc, and vitamins (folic acid, vitamin C, and vitamin B12) [6]. In contrast, excesses of sucrose, sodium, total fats, saturated fatty acids, and cholesterol have been noted [6]. With the aim of cardiometabolic prevention, women with PCOS are advised to increase their intake of plant-based foods—including whole grains, fruits, vegetables, legumes, nuts, and less- processed foods of animal origin [7]. If insulin resistance is present, they should also lower the glycaemic index and saturated fats of food they consume. Our previous research has shown that women with PCOS consume fewer low glycaemic index foods than those in a control group [8]. Additionally, women who restrict sugar (foods with a high glycaemic index) tend to have better diet quality overall than women who do not [9]. Some studies indicate that vitamin B3 supplementation could be valuable, as a deficiency in vitamin B3 contributes to inflammation and thus increases cardiometabolic risk [10,11]. While drawing attention to cardiometabolic risk protection in women with PCOS, it has recently been shown that 8 weeks of CoQ10 supplementation had a beneficial effect on inflammatory and endothelial dysfunction markers in overweight and obese patients with PCOS [12,13,14]. In addition, vitamin D supplementation increases insulin synthesis, insulin receptor expression, and insulin response to glucose transport [15]. Similarly, herbal supplements seem to be highly effective in combating chronic inflammation (Curcuma longa) and improving liver steatosis (Silybum marianum, Nigella sativa) in PCOS [16,17].

Recently, the relationship between the gut-brain axis and health has been discussed. Multiple studies suggest using probiotics to manage metabolic syndrome; however, results vary [18,19,20]. *Lactobacillus* and *Bifidobacterium*, found in naturally fermented foods such as yoghurt, could improve the gut microbiota of obese patients and therefore improve metabolic syndrome status [21]. Some studies suggest that probiotic supplementation in both humans and animals lowers plasma’s blood lipid and atherogenic index, a strong marker of coronary artery disease [22,23,24]. Additionally, it has been shown that microbiota is affected by BMI and body composition [25,26]. After weight reduction in obese patients, the total abundance of bacteria increases; specifically, the ratio of *Firmicutes/Bacteroidetes* decreases, and *Lactobacilli* significantly increase [25].

The novel concept of “microgenderome,” the potential interaction between sex hormones and gut microbiota, has also recently emerged in microbiota research [27]. This theory suggests that the composition of the commensal microbiome of males and females becomes different during puberty, and that sex hormone levels have specific effects on microbiota composition. A mouse model found that a decrease in gut microbiota increased testosterone concentration in female mice but decreased it in male mice. Thus, the commensal gut microbiota affected the production of the male sex hormone [27,28]. Taken together, these results and the research on intestinal microbiota problems in women with PCOS indicate that this relationship needs to be investigated; some researchers suggest that the treatment of PCOS should include probiotic supplementation [29]. Additionally, research shows that the intake of polyphenols from whole grains can regulate both microbiota and serum lipid profile [30]. Therefore, considering that women with PCOS have microbiota alterations [28], the intake of prebiotics and high-fibre foods should be emphasised, and recommendations for consumption should be discussed.

Although cardiometabolic risk can be life-threatening in women with PCOS, research on the consumption of synbiotic foods in this population is limited, so studies should be developed with this aim. It is worth noting that the search for markers that estimate cardiometabolic risk in different populations, including women with PCOS, is widely discussed in the literature [3,31]. However, few studies have reported on the properties of dietary probiotics that lead to changes in cardiometabolic markers in women with PCOS. Among several such markers, the atherogenic index of plasma (AIP) is a logarithmic transformation of the ratio of triglycerides and high-density lipoproteins [24]. AIP is inversely related to LDL particle size, and small dense LDL-C is very vulnerable to oxidative damage; therefore, it is more likely to cause atherosclerotic lesions. This marker is considered a good early predictor of cardiovascular disease in women with PCOS and has previously been used in research with this population [3,18,32,33,34]. Nevertheless, AIP has never been analysed in relation to the intake frequency of antiatherogenic foods.

Despite the apparent benefits of synbiotic foods on health outcomes, their effect on PCOS outcomes remains unclear. Additionally, the frequency of consumption of selected food groups that guaranteed improvement was not identified. Those facts prompted us to look for associations between atherosclerosis risk and the consumption of synbiotic food groups. With this in mind, we analysed the relationship between the consumption frequency of pro- and prebiotic foods and atherosclerosis risk, as such consumption could have a potential protective effect in supporting the treatment of women with PCOS.

## 2. Materials and Methods

### 2.1. Study Participants

A total of 127 women of reproductive age diagnosed with PCOS were recruited from the Department of Endocrinology, Metabolism and Internal Medicine, Poznan University of Medical Sciences (Figure 1). Participants were classified according to Rotterdam criteria. Specific inclusion criteria for the study group were as follows: PCOS diagnosis (according to Rotterdam criteria), age 18–40, and BMI 18–35 kg/m^2^. Exclusion criteria for both low and medium-high AIP groups comprised: chronic hepatic, renal or rheumatic diseases; overt hypothyroidism; and pregnancy. Written informed consent was obtained from all participants. The clinical examination protocol complied with the Declaration of Helsinki for Human and Animal Rights and its later amendments and received ethical approval from the Board of Bioethics of the University of Medical Science (552/16; 986/17).

### 2.2. Atherogenic Markers

Total cholesterol (TC-C), high-density lipoprotein cholesterol (HDL-C), and triglycerides (TG-C) were evaluated using the enzymatic colourimetric method. The Friedewald formula was used to estimate low-density lipoprotein cholesterol (LDL-C). Serum glucose was assessed with the hexokinase method (Roche Diagnostics) and a coefficient of variation (CV) of 3%. The following formula calculated the homeostasis model assessment for insulin resistance (HOMA-IR): HOMA-IR = (fasting plasma glucose (mg/dL) × fasting plasma insulin (mU/L))/405. To determine IR, a threshold of HOMA-IR > 2.5 was used [35]. The atherogenic index of plasma (AIP) was calculated as the logarithm of the total triglycerides to high-density lipoproteins ratio. A ratio of 0.11 and above was considered medium–high risk [36].

### 2.3. Food Frequency Intake and Diet Quality Indexes

Food frequency intake was assessed using the validated Dietary Habits and Nutrition Beliefs KomPAN^®^ Questionnaire, which consists of 25 questions about different food groups. According to the manual, each answer is rated from 0 (never eaten) to 2 (eaten twice a day or more) [37,38]. For each food item, the frequency consumption categories were converted to values reflecting daily consumption (1) never = 0.00, (2) 1–2 times/month = 0.06, (3) once a week = 0.14, (4) 2–3 times/week = 0.5, (5) once a day = 1 and (6) a few times during the day = 2. These conversions are accepted in the literature and bear out the authors’ own experiences [39,40].

The intensity of food frequency intake was characterised using diet indexes for products widely considered healthy, unhealthy, probiotic, and low glycaemic index To evaluate overall diet quality, pro-Healthy-Diet-Index (pHDI-10), non-Healthy-Diet-Index (nHDI-14), Probiotic diet index (ProDI-4), and low-glycaemic diet index (lGIDI-4) scores were established based on previous knowledge and other validated studies [8,41].

Each index was designed and calculated based on the same conversion formula as previously validated in [8,38]: Groups of food products classified in the individual diet indexes are presented in Table 1.
$$Diet\;Index\;\left(\%\right)=\frac{\sum \;A*100\%}{\sum \;B}$$
where *A* was the patient’s actual frequency intake per day, and *B* was the patient’s maximum possible frequency intake per day. For example, for the probiotic food index:
∑*A* = 0.06 + 0.14 + 1 + 2 and ∑*B* = 2 + 2 + 2 + 2.

For each food group (diet index), consumption level was classified on a percentage scale. Each diet index was expressed in % points and was categorised as follows: low- (0–33.32% points), moderate- (33.33–66.65% points), or high- (66.66–100% points) intensity consumption of selected food groups.

To control for both fat intake and for preferences of whole fat, low fat, and non-fat dairy, as well as for sweetened and non-sweetened diary, the following questions were added to the analysis:What is the usual fat content of dairy products you consume?Do you consume sweetened milk drinks and desserts for snacks?Do you consume non-sweetened milk drinks and desserts for snacks?

The answers to these questions were analysed in Table 2.

### 2.4. Body Composition

Patient body composition was assessed using air plethysmography via BodPod (Life Measurement Inc., Concord, CA, USA); measurements were performed according to the validated protocol. Participants came to sessions after overnight fasting. The patients were advised to wear approved clothing, such as bathing suits, compression shorts, and bras with no wiring or padding, and not to wear any jewellery. Additionally, each patient wore a swim cap to decrease hair volume. The equipment was calibrated every morning before study sessions; each session took place at 21–26 °C with relative humidity between 20–70%. Patients’ height, weight, and waist and hip circumferences were also recorded. Measurements (and their respective cut-off points) were performed according to WHO recommendations [42] (Appendix A)

### 2.5. Statistics

Statistical analysis was performed using Statistica (Stat Soft, Krakow, Poland) software. Differences in patient characteristics between the two groups were calculated using independent samples t-tests. For data that was not normally distributed, the Mann-Whitney U test was used. Chi-square tests were performed for non-parametric and categorisation data. Logistic regression analysis was used to estimate odds ratios (OR) and 95% confidence intervals (95% CI) of the estimated dietetic intake in relation to the atherogenic index of plasma. Statistical analyses were performed according to other previously published studies [8,41,43].

## 3. Results

A total of 127 women with PCOS took part in the study (Figure 1). Women were divided into two groups: one of the participants with either a moderate or high risk of atherosclerosis (AIP ≥ 0.11; n = 59), and one of those with a low risk of atherosclerosis (AIP < 0.11; n = 68). Women with high AIP had significantly higher metabolic markers (body mass, BMI, waist circumference, WHR, total cholesterol, LDL, triglycerides, HOMA, fasting insulin, and glucose). HDL was significantly higher in women with low API, and women with high API tended to have a higher fat percentage than women with low API. Fat-free mass and age were not significantly different between the two groups (Table 2).

In both groups, low-intensity consumption of both pre-and probiotic foods was observed; we noted values below 33%. Nevertheless, in the low AIP group, preDI-4 was significantly lower than in the group with high AIP (30% vs. 25%; *p* < 0.05). There was also no significant difference in consumption of sweetened milk drinks and desserts, unsweetened milk drinks and desserts, and low and whole-fat milk between groups. Additionally, 13% of the women did not consume dairy at all.

The association between food frequency intake and high AIP is shown in Table 3. Women with AIP equal to or above 0.11 were 68% less likely to have lGIDI-4 equal to or above the upper quartile than women with low AIP (Figure 2, Appendix A). Women with high AIP were also 65% less likely to consume buckwheat, oats, whole-grain pasta or coarse-ground grains at least a few times a week (Table 3).

## 4. Discussion

The present study is an original investigation examining the effect of synbiotic food consumption on atherosclerosis risk in women diagnosed with PCOS. Overall, we observed that consuming low glycaemic index foods at least twice a week decreased cardiometabolic risk.

Before discussing prebiotic, probiotic or synbiotic foods, it is essential to clarify that their health-promoting properties are widely confirmed [44,45,46]. Probiotics are living microorganisms that, when administered adequately, confer a health benefit to the host [47,48]. Prebiotics are non-digestible foods that positively affect the growth and/or activity of certain bacteria in the gastrointestinal tract, thus improving the host’s condition [49]. Synbiotics combine probiotics and prebiotics; all can be found in popular food products. Prebiotics are found in wholemeal grains, chicory root, garlic, onions, legumes, or apples; the soluble fibre present in those foods feeds beneficial microbiota in the human large intestine. Additionally, reports have shown that the intake of polyphenols influences the gut-brain axis [50]. Probiotics can be found in fermented foods like yoghurt, kimchi, pickles, or tempeh.

Since eating behaviour is complex and respondents eat foods with similar properties; we designed probiotic and low glycaemic, high-fibre diet indexes. The probiotic index included fermented milk drinks, tinned vegetables, fresh cheese-curd products, and cheese. In comparison, the low glycaemic index consisted of wholemeal bread, buckwheat, oats, whole-grain pasta or other coarse-ground grains, legume-based foods, and vegetables.

Compared with women with PCOS and high AIP, those with low AIP reported higher-intensity consumption of low glycaemic index (prebiotic) foods, like wholemeal bread, buckwheat, oats, whole-grain pasta or other coarse-ground groats, legume-based foods, and vegetables. A high frequency of consumption of products from this particular group (above the top quartile: 38.25%) indicated a significantly lower (68%) cardiometabolic risk. However, the risk changed when we excluded legumes and vegetables from this group. It turned out that they did not have as strong a protective effect as wholemeal bread, buckwheat, oats, whole-grain pasta or other coarse-ground groats. Ultimately, the analysis showed that consuming products from this particular group at least twice a week reduced cardiometabolic risk by 65%.

Given the discussion in the literature concerning carbohydrate intake, these results may be surprising. In recent years, the ketogenic diet has gained popularity because of its low glycaemic index; the ketogenic diet is characterised by very restrictive carbohydrate consumption. Some studies show that this diet positively influences insulin sensitivity and lipoprotein and androgen status in women with PCOS [51,52,53]. However, these studies did not include control groups, so a broad conclusion is difficult. Other studies show that both a high intake of whole-grain carbohydrates and a low glycaemic index effectively support pharmacological therapy for PCOS [54,55,56,57]. In our study, the decisive role in decreasing atherosclerosis risk was played by low glycaemic index products, which are a significant source of carbohydrates with prebiotics, soluble fibre, and antioxidant.. The diet’s antioxidant capacity is hard to ignore when discussing the beneficial effects of its nutrition; our previous studies have repeatedly highlighted and discussed this [58,59].

The current results have unique significance because of the combination of nutritional intake analysis and atherosclerosis risk. To examine this, we chose markers like body fat composition and anthropometrics (Appendix A), fasting glucose and insulin, lipid profile, HOMA-IR, and atherogenic index of plasma (AIP). AIP was used to calculate atherosclerosis risk; its use in women with PCOS has been previously demonstrated in the literature [60]. The AIP is based on a positive association with lipoprotein particle size, cholesterol esterification rates, and remnant lipoproteinemia, and it is even recommended as a sole marker of cardiovascular diseases (CVD) [61]. AIP values of −0.3 to 0.1 are associated with a low risk of CVD, values of 0.1 to 0.24 with a medium risk, and values above 0.24 with a high risk [36].

We must emphasise that our results have important implications for managing dietary recommendations for women with PCOS. In our study, we analysed the consumption of all food groups; nevertheless, only low glycaemic index products affected atherosclerosis risk as expressed by the atherogenic index of plasma (AIP). In that group, the consumption of buckwheat, oats, whole-grain pasta or other coarse-ground groats strongly influenced atherosclerosis risk. This outcome supports other studies concluding that the quality of consumed carbohydrates influences atherosclerosis risk. However, we did not find an adverse relationship between AIP and low-quality carbohydrates (processed grains or white bread). Notably, the groups we found significant included products containing high amounts of soluble fibre. Soluble fibre feeds probiotic bacteria and decreases cholesterol by binding bile salts to the intestinal passage and excreting it with faeces [62,63]. In total, 10 pro-healthy food groups had a tendency to lower AIP.

Although the beneficial effect of probiotics on atherosclerosis has been previously observed in the literature [18,63], our research quantifies this relationship for the first time. Interestingly, we found no correlation between consumption of probiotic foods and AIP; probiotic foods were mainly represented by dairy products. Frequent dairy intake seems to lower the risk of diabetes mellitus; however, it is unclear whether it benefits women with PCOS [64,65]. It is also known that low-fat, unsweetened dairy intake has much better health outcomes than sweetened, full-fat dairy intake; low-fat dairy is recommended to reduce the risk of atherosclerosis [7]. In our study, there was no significant association between the intake of sweetened dairy and the risk of atherosclerosis. A high risk of atherosclerosis was also not significantly associated with whole-fat dairy intake. This result agrees with research showing that including whole-fat diary in an otherwise healthy dietary pattern is not associated with hypercholesterolemia [66].

Despite reaching its aims, this study may have some limitations. For example, the calculation of both prebiotics and probiotics is limited; specifically, we used the low glycaemic index dietary score to assess prebiotic intake. Although this index includes most prebiotic foods (such as breads, grains, vegetables, and legumes), it does not include single products from different groups (e.g., apples or onions, which are also high-prebiotic foods). However, using a strong atherosclerosis marker, a validated food frequency questionnaire appropriate for the Polish population, and a thorough body composition analysis, this study provides valuable information on maintaining cardiovascular health in women with PCOS. It should be emphasised that the obtained results could be supported by microbiota analysis (currently a missing element), as this could provide a complete picture of microbiota in women with PCOS. Following this analysis, further research should include a high-fibre diet intervention study. Although we have observed the associations between dietary patterns and atherosclerosis risk in women with PCOS, the mechanisms behind it are still unclear.

## 5. Conclusions

In conclusion, intake of prebiotic foods is inversely associated with a medium or high risk of atherosclerosis in women with PCOS. Such an association has not been found between atherosclerosis risk and probiotic foods. Our data provide further evidence that promoting dietary recommendations to consume good quality carbohydrates and low GI products in a balanced diet should be considered part of a PCOS treatment plan. Future multidisciplinary approaches involving dietary intervention and microbiota analysis should be regarded to characterise changes meant to counteract atherosclerosis risk in women with PCOS. We believe that further research should move in this direction.

## Figures and Tables

**Figure 1 life-13-00799-f001:**
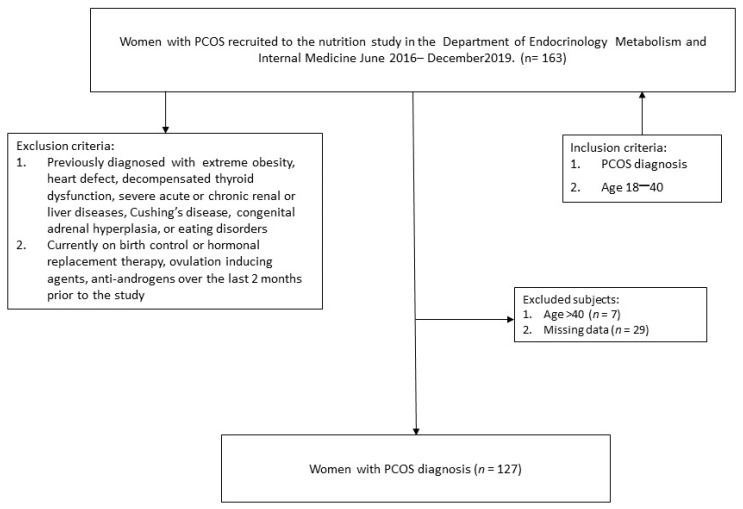
Patient recruitment flowchart.

**Figure 2 life-13-00799-f002:**
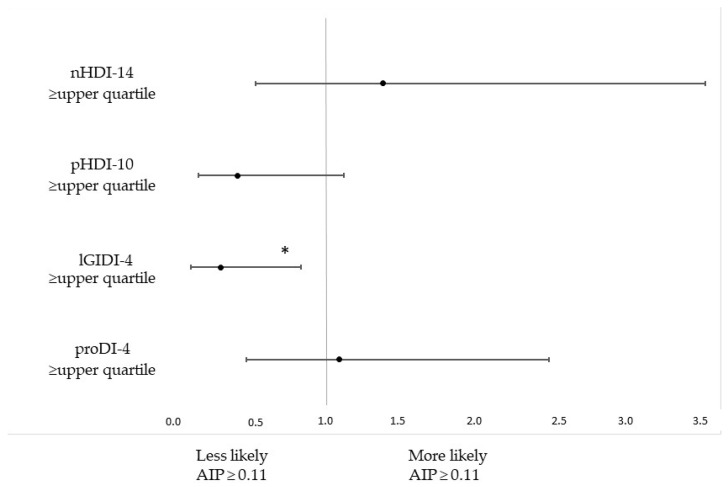
Odds ratios with 95% confidence intervals for upper quartiles of diet quality indexes when the atherogenic index of plasma (AIP) was above or equal to 0.11, adjusted for age and BMI. *p* values below the statistical significance threshold are marked with * (*p* < 0.05).

**Table 1 life-13-00799-t001:** Probiotic (ProDI-4), low glycaemic (lGIDI-4), pro-healthy (pHDI-10), and non-healthy (nHDI-14) diet quality indexes and their product content.

Food Group	Products Included
proDI-4 ^1^	(1) Fermented milk drinks, (2) Pickled vegetables, (3) Fresh cheese-curd products, (4) Cheese
lGIDI-4 ^2^	(1) Wholemeal bread, (2) Buckwheat, oats, whole-grain pasta or other coarse-ground grains, (3) Legume-based foods, (4) Vegetables
pHDI-10 ^3^	(1) Wholemeal bread, (2) Buckwheat, oats, whole-wheat pasta, (3) Milk, (4) Fermented milk drinks, (5) Fresh cheese-curd products, (6) White meat, (7) Fish, (8) Legume-based foods, (9) Fruits, (10) Vegetables
nHDI-14 ^4^	(1) White bread, (2) White rice, pasta, fine-ground grains (3) Fast food, (4) Fried dishes, (5) Butter, (6) Lard, (7) Cheese, (8) Cold meats, smoked sausages, hot dogs, (9) Red meat dishes, (10) Sweets, (11) Tinned meats, (12) Sweetened carbonated and non-carbonated drinks, (13) Energy drinks, (14) Alcoholic beverages

^1^ Probiotic diet index, ^2^ low glycaemic diet index, ^3^ Pro-healthy diet index, ^4^ Non-healthy diet index.

**Table 2 life-13-00799-t002:** Characteristics of the study sample and their dairy preferences.

Variable	Low AIP (n = 68)		High AIP (n = 59)		
	Mean ± SD	Median (CI95%)	Mean ± SD	Median (CI95%)	*p*
Age (y)	26 ± 5	25 (4; 6)	26 ± 6	25 (5; 7)	0.74
Body mass (kg)	66.4 ± 11.2	63.6 (11.2; 9.6)	75.3 ± 16.1	72.7 (13.6; 19.7)	0.00 *
BMI (kg/m^2^)	23.5 ± 3.3	22.9 (2.9; 4.0)	27.4 ± 5.7	26.8 (4.8; 7.0)	0.00 *
Waist circumference (cm)	77.3 ± 9.9	76.0 (8.4; 11.9)	86.5 ± 14.3	88.0 (12; 18)	0.00 *
WHR (-)	0.79 ± 0.06	0.78 (0.05; 0.07)	0.84 ± 0.10	0.80 (0.09; 0.12)	0.00 *
FM (%)	37.2 ± 22.9	33.0 (19.5; 27.7)	45.8 ± 25.8	41.1 (21.8; 31.6)	0.05
FFM (%)	70.5 ± 16.9	68.2 (14.4; 20.5)	66.9 ± 21.2	60.5 (17.9; 26.1)	0.30
T. Cholesterol (mg/dL)	170.6 ± 29.2	168.5 (25.0; 35.2)	187.4 ± 32.0	185.0 (27.1; 39.2)	0.00 *
HDL (mg/dL)	75 ± 14	73.0 (12.0; 16.9)	55 ± 11	55.0 (9.0; 13.0)	0.00 *
LDL (mg/dL)	84 ± 25	83.0 (21.5; 30.3)	108 ± 30	110.4 (25.7; 37.1)	0.00 *
TG (mg/dL)	55 ± 15	53.0 (12.7; 17.9)	123 ± 67	103 (56; 81)	0.00 *
Fasting glucose (mg/dL)	87 ± 7	88 (6; 9)	91 ± 7	90 (5.6; 8.0)	0.00 *
Fasting insulin (uU/mL)	8.10 ± 3.63	7.35 (3.10; 4.37)	14.59 ± 8.58	12.4 (7.3; 10.5	0.00 *
HOMA-IR (-)	1.79 ± 0.89	1.58 (0.76; 1.07)	3.37 ± 2.19	2.9 (1.8; 2.7)	0.00 *
AIP ^1^ (-)	−0.33 ± 0.29	−0.35 (0.25; 0.36)	0.73 ± 0.55	0.59 (0.47; 0.67)	0.00 *
proDI-4 ^2^ (%)	13.85 ± 8.75	14.12 (7.48; 10.53)	14.84 ± 8.26	14.00 (6.99; 10.10)	0.52
lGIDI-4 ^3^ (%)	30.47 ± 16.19	27.50 (13.85; 19.48)	25.27 ± 14.29	26.75 (12.09; 17.46)	0.05 *
pHDI-10 ^4^ (%)	27.50 ± 11.04	26.10 (9.44; 13.29)	23.99 ± 10.72	20.3 (9.07; 13.10)	0.07
nHDI-14 ^5^ (%)	13.63 ± 7.83	11.39 (6.70; 9.43)	13.92 ± 8.03	11.86 (6.80; 9.82)	0.83
	n	%	n	%	
Unsweetened milk drinks as a snack	25	20	21	17	0.89
Sweetened milk drinks as a snack	12	9	18	14	0.08
Milk and milk drinks					0.68
Whole fat	27	21	27	21	
Low fat	30	24	27	21	
Non-fat	2	2	1	1	
No dairy	9	7	4	3	

^1^ AIP—atherogenic index of plasma; ^2^ proDI-4—probiotic diet index; ^3^ lGIDI-4—low glycaemicdiet index; ^4^ pHDI-10—pro-healthy diet index; ^5^ nHDI-14—non-healthy diet index. *p* values below the statistical significance threshold are marked with * (*p* < 0.05). Non-parametric values were calculated with chi-square tests.

**Table 3 life-13-00799-t003:** Odds ratios (ORs with 95% confidence intervals (95% CI)) of the high atherogenic index of plasma according to the consumption frequency of selected food groups.

Food Groups	Atherogenic Index of Plasma ≥ 0.11		
	Occurrence (%)/n	Crude OR (CI 95%)	OR Adjusted for BMI and Age (CI 95%)
Wholemeal (brown) bread/bread rolls			
≥once a day	(13)/16	1.21 (0.54; 2.72); *p* = 0.64	1.18 (0.48; 2.89); *p* = 0.71
Buckwheat, oats, whole-grain pasta or other coarse-ground grains			
≥two times a week	(16)/20	0.38 (0.18; 0.79); *p* = 0.01 *	0.35 (0.16; 0.78); *p* = 0.01 *
Pickled vegetables			
≥two times a week	(7)/9	1.35 (0.48; 3.80); *p* = 0.57	0.82 (0.24; 2.84); *p* = 0.76
Milk (including flavoured milk, hot chocolate, or latte)			
≥once a day	(16)/19	0.64 (0.31; 1.33); *p* = 0.23	0.51 (0.22; 1.18); *p* = 0.11
Fermented milk drinks, e.g., yoghurts, kefir (natural or flavoured)			
≥once a day	(8)/10	1.18 (0.45; 3.11); *p* = 0.73	0.96 (0.32; 2.85); *p* = 0.94
Fresh cheese-curd products, e.g., cottage cheese, cream cheese, cheese-based puddings			
≥once a day	(2)/2	1.16 (0.16; 8.66); *p* = 0.86	1.03 (0.10; 10.4); *p* = 0.98
White meat, e.g., chicken, turkey, rabbit			
≥two times a week	(35)/45	1.25 (0.56; 2.80); *p* = 0.59	1.25 (0.56; 2.80); *p* = 0.59
Fish			
≥two times a week	(4)/5	0.61 (0.19; 1.95); *p* = 0.40	0.47 (0.13; 1.78); *p* = 0.26
Legume-based foods, e.g., beans, peas, soybeans, lentils			
≥once a week	(16)/19	0.72 (0.35; 1.51); *p* = 0.38	0.60 (0.26; 1.37); *p* = 0.22
≥two times a week	(6)/7	0.60 (0.21; 1.55); *p* = 0.27	0.50 (0.16; 1.54); *p* = 0.22
Fruits			
≥two times a day	(9)/12	0.66 (0.29; 1.52); *p* = 0.32	0.59 (0.23; 1.51); *p* = 0.27
Vegetables			
≥two times a day	(17)/22	1.16 (0.56; 2.43); *p* = 0.68	1.21 (0.54; 2.70); *p* = 0.64
White bread and bakery products, e.g., wheat bread, rye bread, wheat-rye bread, toast bread, bread rolls			
≥once a day	(14)/18	0.71 (0.34; 1.50); *p* = 0.36	0.98 (0.43; 2.26); *p* = 0.97
White rice, white pasta, fine-ground groats, e.g., semolina, couscous			
≥two times a week	(12)/15	0.55 (0.26; 1.19); *p* = 0.13	0.67 (0.29; 1.56); *p* = 0.35
Fast foods, e.g., potato chips/French fries, hamburgers, pizza, hot dogs			
≥once a week	(9)/11	0.81 (0.34; 1.95); *p* = 0.64	1.03 (0.40; 2.67); *p* = 0.95
Fried foods (e.g., meat or flour-based foods such as dumplings, pancakes, etc.)			
≥two times a week	(20)/26	1.12 (0.52; 2.43); *p* = 0.77	1.12 (0.52; 2.43); *p* = 0.77
Butter as a bread spread or as an addition to your meals/for frying/for baking, etc.			
≥once a day	(15)/19	1.06 (0.50; 2.27); *p* = 0.87	0.85 (0.36; 1.98); *p* = 0.70
Lard as a bread spread, or as an addition to your meals/for frying/for baking, etc.			
≥once a week	(1)/1	0.28 (0.03; 2.60); *p* = 0.26	0.25 (0.02; 3.09); *p* = 0.27
Cheese (including processed cheese and blue cheese)			
≥two times a week	(24)/30	1.10 (0.54; 2.22); *p* = 0.80	1.34 (0.61; 2.94); *p* = 0.46
Cured meat, smoked sausages, hot dogs			
≥two times a week	(26)/33	1.35 (0.66; 2.73); *p* = 0.41	1.45 (0.66; 3.17); *p* = 0.35
Red meat, e.g., pork, beef, veal, lamb, game			
≥two times a week	(16)/20	1.98 (0.88; 4.43); *p* = 0.10	1.81 (0.75; 4.36); *p* = 0.18
Sweets, e.g., confectionery, biscuits, cakes, chocolate bars, cereal bars, etc.			
≥two times a week	(31)/39	1.06 (0.51; 2.23); *p* = 0.87	1.25 (0.55; 2.85); *p* = 0.59
Tinned (jar) meats			
≥1–3 times a month	(9)/12	1.32 (0.53; 3.30); *p* = 0.54	0.94 (0.33; 2.67); *p* = 0.91
Sweetened carbonated or still drinks			
≥two times a week	(6)/8	1.18 (0.41; 3.39); *p* = 0.76	0.92 (0.27; 3.08); *p* = 0.89
Energy drinks			
≥once a week	(2)/2	0.76 (0.12; 4.80); *p* = 0.77	1.01 (0.14; 7.61); *p* = 0.99
Alcoholic beverages			
≥once a week	(10)/13	0.79 (0.34; 1.79); *p* = 0.56	0.79 (0.32; 1.97); *p* = 0.61

*p* values below the threshold of statistical significance are marked with * (*p* < 0.05).

## Data Availability

The data supporting the conclusions of this article are included within the article and its additional files. The other datasets used and/or analysed during the current study are available from the corresponding author upon reasonable request.

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
