# Peer review of "Intake of Low Glycaemic Index Foods but Not Probiotics Is Associated with Atherosclerosis Risk in Women with Polycystic Ovary Syndrome"

_life, 2023, doi:10.3390/life13030799_

Round 1

Reviewer 1 Report

The study aims to analyse the relationship between the frequency of dietary intake of pro- and prebiotic foods and atherosclerosis risk in women with PCOS. It was quite well-design study, but I have a few comments to improve the manuscript:

Title: I propose a statement instead of the question

Introduction: add information how BMI and body composition affect the microbiome;  the second paragraph is too long

Materials and Methods: add references for Rotterdam criteria and whether the women included in this study were being treated for metabolic disorders or were on any diet. I don't understand why 273 women were classified with PCOS and then excluded 110, when one of the inclusion criteria was PSOC diagnosis (fig. 1)

line 147 - add more references

Table 1. - which "tinned vegetables" were clasified as probiotic food?

2.4 and table 2. - what body size measurements were taken? How BMI, WHR was calculated? What cut-points were used? Results and data analysis should be supplemented with the obtained anthropometric data

line 216 -I couldn't find Supplement 1

Table 2 - please rewrite mean ± SD and add median and range. Add  BMI, FM, WC and metabolic markers distribution for AIP groups. Which data was tested at U Mann Whitney? In 2.5 chi2 -it wasn't mentioned

line 225 -Table 3 not 2

Table 3.(now 2) -  why OR was adjusted only for BMI and age?

Discussion - should be supplemented with the obtained anthropometric data in association to AIP and foods indexes.

lines 312-317 -these are not limitations for your study. Now all refer to future research. Add also strenght.

Minor comments:

- you used twice "food ingredients' - in what sense?

- synbiotic no symbiotic

- line 232 -  you didn't define "synbiotic oriented dietary patterns" - it is overinterpretation

- line 245 - delete "Probiotic"

Author Response

Dear Reviewers,

Thank you for revising our manuscript titled: Intake of low glycaemic index foods but not probiotics is asso-ciated with atherosclerosis risk in women with polycystic ovary syndrome

We greatly appreciate the time and efforts to review our manuscript and agree that the proposed changes will contribute to improving our manuscript. We have addressed all issues indicated in the review, and we believe that the revised version can meet the journal publication requirements.

Please find our responses attached in the document.

Yours Sincerely,

Magdalena Czlapka-Matyasik

Reviewer 2 Report

Overall, this is a well-written manuscript. The author showed a very detailed introduction. The data used was nicely depicted. The descriptive statistics was neatly presented and easy to understand. The logit regression model result was also well presented, however, the authors didn't check the residuals plot to see how the data points distributed.  The discussion and conclusion parts were nicely presented, I don't have any concerns. 

Author Response

Dear Reviewer,

Thank you very much for the positive revision. We added the median and confidence intervals for each food group analysed. We hope, that this will sufficiently present the distribution of our data. We greatly appreciate the time and efforts to review our manuscript.

Best regards,

Magdalena Czlapka-Matyasik

Reviewer 3 Report

This study aims to analyze the relationship between the frequency of dietary intake of pro- and prebiotic foods and atherosclerosis risk in women with polycystic ovary syndrome (PCOS). The final results shown that women with PCOS and at high risk of atherosclerosis consumed less prebiotic foods than women with a low risk of atherosclerosis. Therefore, it thought high prebiotic foods could prevent atherosclerosis in women with PCOS. At present, a growing number of studies have proved probiotics and prebiotic foods can prevent disease (such as cardiovascular disease, hypercholesterolemia and so on). It is a greatly meaningful to carry out the study for women with PCOS.    

Generally speaking, the manuscript is well organized, which can be published in the journal. However, there still have some issues need to be revised in the manuscript.

Introduction

1.      Line 58: “Lactobacillus and Bifidobacterium found in naturally fermented foods, such as yoghurt, could improve the microbiota in the gut of obese patients and therefore improve metabolic syndrome status”. The bacteria names are suggested to be italicized.

2.      Line 59-61 “Summing up the literature reports, it should be emphasised that often, the recommendations are for single ingredients or supplements. Recommendations referring to general behaviours, food group consumption or frequency of intake are limited” In fact, the oat or polyphenols in wholegrain can regulate the gut microbiota and serum lipid(Food & Function, 2022, 13(24), 12686-12696. Doi: 10.1039/d2fo01746f. ).

3.      Line67-68. “Multiple studies suggest using probiotics to manage metabolic syndrome; however, the results vary”. There is a positive correlation for the polyphenols and prebiotic effect.

Materials and methods

1.Line 123: “The Friedewald formula was used to estimate Low-density lipoprotein cholesterol (LDL-C) was calculated.” Whether the sentence is problematic?

Results

1.Line 201: “In both groups, low-intensity consumption of pre-, probiotic foods was observed; we noted values below 33%.……” has different font sizes.

2.The Tables are suggested to be rearrange more excellent.

3. The format layout of this paper requires modification. For instance, line 211 and 214. 

4. Line 237-238. The food consumption or prebiotic factors should be reconsidered (Critical Reviews in Food Science and Nutrition. Doi: 10.1080/10408398.2022.2076064).

Author Response

Dear Reviewer,

Thank you for revising our manuscript titled: Intake of low glycaemic index foods but not probiotics is asso-ciated with atherosclerosis risk in women with polycystic ovary syndrome

We greatly appreciate the time and efforts to review our manuscript and agree that the proposed changes will contribute to improving our manuscript. We have addressed all issues indicated in the review, and we believe that the revised version can meet the journal publication requirements.

Please find our responses attached in the document.

Yours Sincerely,

Magdalena Czlapka-Matyasik

Round 2

Reviewer 3 Report

The authors of this manuscript has responsed the reviewer's comments point by point. It can be accepted in current revision.